# Scan-Free GEXRF in the Soft X-ray Range for the Investigation of Structured Nanosamples

**DOI:** 10.3390/nano12213766

**Published:** 2022-10-26

**Authors:** Steffen Staeck, Anna Andrle, Philipp Hönicke, Jonas Baumann, Daniel Grötzsch, Jan Weser, Gesa Goetzke, Adrian Jonas, Yves Kayser, Frank Förste, Ioanna Mantouvalou, Jens Viefhaus, Victor Soltwisch, Holger Stiel, Burkhard Beckhoff, Birgit Kanngießer

**Affiliations:** 1TU Berlin, Analytical X-ray Physics, 10623 Berlin, Germany; 2Physikalisch-Technische Bundesanstalt, 10587 Berlin, Germany; 3Helmholtz-Zentrum Berlin, 12489 Berlin, Germany; 4Max-Born-Institut, 12489 Berlin, Germany

**Keywords:** GEXRF, periodic nanostructures, soft X-ray

## Abstract

Scan-free grazing-emission X-ray fluorescence spectroscopy (GEXRF) is an established technique for the investigation of the elemental depth-profiles of various samples. Recently it has been applied to investigating structured nanosamples in the tender X-ray range. However, lighter elements such as oxygen, nitrogen or carbon cannot be efficiently investigated in this energy range, because of the ineffective excitation. Moreover, common CCD detectors are not able to discriminate between fluorescence lines below 1 keV. Oxygen and nitrogen are important components of insulation and passivation layers, for example, in silicon oxide or silicon nitride. In this work, scan-free GEXRF is applied in proof-of-concept measurements for the investigation of lateral ordered 2D nanostructures in the soft X-ray range. The sample investigated is a ${\mathrm{Si}}_{3}N_{4}$ lamellar grating, which represents 2D periodic nanostructures as used in the semiconductor industry. The emerging two-dimensional fluorescence patterns are recorded with a CMOS detector. To this end, energy-dispersive spectra are obtained via single-photon event evaluation. In this way, spatial and therefore angular information is obtained, while discrimination between different photon energies is enabled. The results are compared to calculations of the sample model performed by a Maxwell solver based on the finite-elements method. A first measurement is carried out at the UE56-2 PGM-2 beamline at the BESSY II synchrotron radiation facility to demonstrate the feasibility of the method in the soft X-ray range. Furthermore, a laser-produced plasma source (LPP) is utilized to investigate the feasibility of this technique in the laboratory. The results from the BESSY II measurements are in good agreement with the simulations and prove the applicability of scan-free GEXRF in the soft X-ray range for quality control and process engineering of 2D nanostructures. The LPP results illustrate the chances and challenges concerning a transfer of the methodology to the laboratory.

## 1. Introduction

The quantitative investigation of periodic nanostructures in the form of transistor architectures [1,2] are of ever-growing importance for the semiconductor industry. This also procures the development of metrological methods to analyze said structures for quality control and process engineering to maintain reasonable production yields. Microscopic techniques such as scanning (SEM) or transmission (TEM) electron microscopy [3] deliver local spatial information at the nanoscale, but either requires extensive sample preparation, is not element-sensitive and consumes the sample. Techniques such as atom probe tomography (APT) [4], secondary ion mass spectroscopy (SIMS) [5] and scanning transmission electron microscopy (STEM) combined with energy-dispersive X-ray fluorescence spectroscopy (EDX) [6] add element sensitivity, but still require sample preparation and are not able to probe larger sample areas for ensemble information. Non-destructive techniques are of special interest, since they allow for additional measurements of the same sample with complementary techniques. Grazing-incidence small-angle X-ray scattering (GISAXS) [7,8] is one example, but it typically lacks optical contrast between different materials. Grazing-incidence X-ray fluorescence spectroscopy (GIXRF) [9,10,11] provides element sensitivity and depth resolution, and it works non-destructively. GIXRF requires a coherent X-ray beam for excitation, so that an X-ray standing wave (XSW) field [12] can emerge. The incoming radiation interferes with itself due to reflection on the sample structure and modulates the fluorescence intensity, which, in case of regular arrays of 2D or 3D nanostructures, leads to a two-dimensional fluorescence interference pattern, depending on the angle with the sample surface and the azimuth angle of the excitation radiation. From the interference pattern, the structural features of interest can be reconstructed [13]. GIXRF requires a parallel beam for spatial coherence and monochromatic radiation for temporal coherence, therefore, it is difficult to apply this technique in the laboratory. Furthermore, the extended footprint at shallow excitation angles restricts the achievable lateral resolution and usually exceeds the size of test fields used in the semiconductor industry. For the routine investigation of nanostructures, more accessible methods are required, which are also applicable in the laboratory.

Grazing-emission X-ray fluorescence spectroscopy (GEXRF) [14,15] bypasses these problems. GEXRF only demands sufficient energy resolution from the detector to differentiate between the fluorescence lines of interest and a sufficient angular resolution to resolve interference features. There are no requirements on the coherence of the excitation radiation, since the interference of the fluorescence emission is recorded, which is inherently temporally coherent due to the narrow natural line widths. The sample may be excited under a 90° excitation angle, which keeps the footprint as small as possible. Since no parallel beam is needed, focusing optics can be employed to reduce the footprint even further, for example, down to 10 μm with capillary optics [16] in the soft X-ray range. In addition to that, the whole setup can be operated scan-free (SF-GEXRF) by using an area detector such as a CCD [17,18] and avoiding to scan the angular range. The application of SF-GEXRF for the investigation of periodic nanostructures has been demonstrated in the tender X-ray range [19]. However, the CCD detector was not able to resolve fluorescence lines in the soft X-ray range to investigate the distribution of light elements such as oxygen, nitrogen or carbon. Oxygen and nitrogen in particular are of importance for the semiconductor industry, since silicon nitride and oxides provide crucial passivation and insulation layers [20]. Further applications of SF-GEXRF include the characterization of multilayers [21], nanoparticles [17] and the depth-resolved chemical speciation (in combination with X-ray absorption spectroscopy) [22]. The application of SF-GEXRF with a complementary technique, for example, X-ray reflectivity (XRR) or SEM measurements, might be possible to enhance the overall sensitivity of the approach.

In this work, the feasibility of the method in the soft X-ray range is demonstrated in proof-of-concept measurements at the BESSY II synchrotron radiation facility, as well as in the laboratory. The superior energy resolution of the CMOS detector compared to common CCD detectors makes SF-GEXRF applicable in the soft X-ray range to investigate lighter elements such as O, N and C [23,24,25]. First, the results of SF-GEXRF measurements of a ${\mathrm{Si}}_{3}N_{4}$ grating at the UE56-2 PGM-2 beamline at the BESSY II synchrotron radiation facility are presented. Additionally, a laser-produced plasma (LPP) source at the Berlin Laboratory for innovative X-ray Technologies (BLiX) [26] is utilized to investigate the potential for routine measurements. The results of both measurements are evaluated by comparing them to calculations performed with a Maxwell solver based on finite elements using a well-known sample structure determined in previous GIXRF measurements [13].

## 2. Materials and Methods

The investigated sample is a silicon nitride lamellar grating on a silicon substrate, which was manufactured by means of electron beam lithography at the Helmholtz-Zentrum Berlin. The nominal pitch is p=100nm, the nominal height h=80nm and the nominal width w=50nm. The sample was manufactured from a 90 nm thick ${\mathrm{Si}}_{3}N_{4}$ layer on a Si substrate. An organic polymer positive resist (ZEP520A) was spin coated onto the substrate. It was developed with a Vistec EBP5000+ e-beam writer with an electron acceleration voltage of 100 kV. Ion etching using ${\mathrm{CHF}}_{4}$ was applied and the remaining resist was removed via oxygen plasma treatment. The structure extents over an area of 1 mm × 15 mm. The sample was already characterized with GIXRF in previous measurements [11]. The sample model from finite-element method (FEM) calculations performed in the process now serves as a reference for the measurements presented [10,11,13]. Oxygen was found on top of the grating structure and between the silicon substrate and the silicon nitride, probably in the form of ${\mathrm{SiO}}_{2}$ [10].

The sample is first investigated in SF-GEXRF geometry at the UE56-2 PGM-2 beamline at the BESSY II synchrotron radiation facility. The beamline uses a plane grating mirror monochromator and the beam size is 1 mm × 1 mm. The detector used is a Tucsen Dhyana 95 CMOS detector [25]. This detector was adapted for the use in the soft X-ray range [27]. For the GEXRF measurements, the sample is aligned at 90° excitation angle and 0° emission angle towards the detector. The low readout noise of the CMOS detector enables sufficient energy resolution in the soft X-ray range below 1 keV by means of single-photon event evaluation [25,28]. The total measurement time is significantly reduced compared to using a conventional CCD by virtually omitting the readout time. The distance between sample and detector of $\left(20.6\pm 0.1\right)$ cm is obtained by comparing the experimental data with the simulation performed for the GEXRF measurement. The excitation energy is set to 690 eV. With an exposure time of 40 ms, about 2100 photons per frame are detected, which translates to approximately 50,000 events per second. In total, 120,000 frames are recorded with a total measurement time of ∼1.5 h. Two hundred dark frames are recorded.

Laboratory proof-of-concept measurements are performed at the BLiX with an LPP as the soft X-ray source. The LPP comprises an Yb:YAG laser with a wavelength of 1030 nm and a repetition rate of 100 Hz, which is used to create a plasma from a Cu target. The laser pulse energy is set to $\left(174\pm 4\right)$ mJ with a pulse length of 1 ns. The highly ionized Cu plasma emits intense fluorescence lines in the soft X-ray range. The radiation at 1078 eV is collected and monochromatized by a pair of highly efficient toroidal multilayer optics [29] and focused to a spot of ∼150 μm full width at half maximum (FWHM). The sample is located in an ultra-high vacuum chamber [30] on a wedge to enable an incident angle of 10°. The exciting radiation is in this way absorbed more easily in the grating structure forming the upper layer of the sample for a more efficient excitation. This causes a theoretical broadening of the excitation spot to 650 μm in the beam direction. The CMOS detector is again positioned in a 90° geometry with respect to the exciting radiation. The angular range covered by the detector equals ∼9°. The distance between sample and detector amounts to $\left(14.1\pm 0.1\right)$ cm as determined with the known interference features from the simulation of the O Kα fluorescence pattern. In total, 80,000 frames with 100 ms exposure time and, subsequently, 345,000 frames with 200 ms exposure time are recorded. This adds up to a total measurement time of ∼21.5 h. On average, ∼470 events per second are detected. Two hundred dark frames for dark frame subtraction are recorded for each of the two measurement series. A schematic view of the laboratory setup is shown in Figure 1.

For all measurements, the detector is cooled down to −15°C and operated in the low-gain high dynamic range (HDR) mode.

For evaluation of the recorded CMOS frames, single-photon events are evaluated using the *clustering* algorithm described in [28]. A compound dark frame is used for background subtraction, and noise thresholds of $T_{1}$ = 6 and $T_{2}$ = 3 are applied (see [28] for details). In this way, both energy and spatial information from the pixels of the photon events are gained. The energy information of all photon events from all frames recorded in one measurement then yields an energy-dispersive spectrum. The spectrum can be analyzed by setting regions of interest (ROIs) for each fluorescence line. Every photon event contained in the respective ROI can be depicted on the detector frame with its spatial information. Then, the intensity pattern for each fluorescence line can be calculated and represented in a so-called photon map. The energy axes of the energy-dispersive spectra shift slightly depending on the number of the detected pixels of the event, causing potential overlap of neighboring fluorescence peaks. To mitigate this, the spectra are calculated separately for events with the same number of contributing pixels. For each of these spectra, the ROI limits for O Kα and N Kα are set manually. The ROI width is reduced so that line overlap is avoided as far as possible, sacrificing counts in the process.

The fluorescence patterns are described in dependence on the angle with the sample surface θ and the azimuth angle φ. An emission angle of $\theta =0{}^{{}^{\circ}}$ describes hypothetical fluorescence emission parallel to the sample surface, accordingly, $\theta =90{}^{{}^{\circ}}$ describes fluorescence emission perpendicular to the sample surface. $\varphi =0{}^{{}^{\circ}}$ denotes the direction parallel to the grating lines of the sample, $\varphi =90{}^{{}^{\circ}}$ denotes the direction perpendicular to the grating lines. Due to the sample symmetry, $\varphi =0{}^{{}^{\circ}}$ also defines the symmetry axis of the fluorescence patterns. The θ-axis is calibrated by comparing features in the measured and simulated photon maps of the O Kα signal. From this calibration, the distance between sample and detector and the tilt of the sample relative to the detector surface are derived. This information together with the CMOS chip geometry allows to calculate the solid angle of detection of each pixel. The calibration regarding φ is performed by identifying the symmetry axis at $\varphi =0{}^{{}^{\circ}}$ manually in the O Kα photon map and by applying the detector-to-sample distance derived from the θ-calibration. With the correct assignment of θ and φ angles to each pixel of the CMOS detector, θ-φ-maps can now be produced by assigning the photon events according to their position on the CMOS frame to an angular grid of 150 × 150 pixels between $0{}^{{}^{\circ}}<\theta <6{}^{{}^{\circ}}$ and $-3{}^{{}^{\circ}}<\varphi <3{}^{{}^{\circ}}$. Thus, each pixel of the experimental θ-φ-maps spans $0.04{}^{{}^{\circ}}$ in both angular directions.

The θ-φ-maps are also normalized to the solid angle of detection of each of their pixels. In addition to the ROI evaluation and to investigate and mitigate the effects of overlap between the N Kα, O Kα and C Kα peaks, local spectra deconvolution is applied. For this, XRF spectra are created by summing up photon events in bins of the size of 20 × 20 CMOS pixels. One bin spans 0.06° in both angular directions. The energy-dispersive XRF spectrum of each bin is then analyzed with the in-house software Specfit [31], which applies background subtraction and peak fitting to gain the net peak intensities, similarly to PyMCA [32]. The Specfit evaluation is only applied for the BESSY II measurement, since the statistics in the laboratory measurement were insufficient.

For analysis, simulations of the fluorescence patterns are conducted. The ${\mathrm{Si}}_{3}N_{4}$ lamellar grating is well known from previous studies and the parameters of the grating are published in [13]. It has been characterized and a validated model exists. A sample schematic is shown in Figure 2.

For the calculations of the fluorescence intensities depending on the X-ray standing wave field, a finite-element method (FEM) is used. The electric field distribution is calculated with a Maxwell solver from JCMwave [33]. The calculation is comparable to the calculation of the GIXRF signal [19], but it needs to be adapted for the fluorescence line energies of O Kα (524 eV) and N Kα (393 eV). The sample parameters used for the simulations shown in this work are the height h=97.7nm, the width of the grating lines w=49.77nm, the sidewall angle $swa=83.54{}^{{}^{\circ}}$, the thickness of the silicon oxide layer on top of the grating lines $t_{t}=2.84\mathrm{nm}$ and the thickness of the silicon oxide layer in the grooves $t_{g}=5.82\mathrm{nm}$. The thickness of the silicon oxide layer between the grating structure and the Si substrate $t_{s}=1.35\mathrm{nm}$ and the pitch p=100nm were kept constant during the optimization process [13]. These parameters may also be gained from fitting the simulation to the experimental data shown in this work. More sample parameters, for example, parameters describing irregularities in the sample structure, can potentially be investigated with an appropriate sample model. The densities of the sample materials can be determined with a reference-free approach [10]. Interference patterns in dependence of the two angles θ and φ (θ-φ-maps, the angular directions are depicted in the side view of the schematic setup in Figure 1) are calculated with an accuracy of $\Delta \theta =0.027{}^{{}^{\circ}}$ within a range from $\theta =0.01{}^{{}^{\circ}}$ to $\theta =8{}^{{}^{\circ}}$ and $\Delta \varphi =0.025{}^{{}^{\circ}}$ from $\varphi =0{}^{{}^{\circ}}$ to $\varphi =2.5{}^{{}^{\circ}}$, leading to a total of 30,401 points of the map for each fluorescence line energy. The simulated fluorescence maps are convolved with a 2D Gaussian with an FWHM of $\Delta \theta =0.014{}^{{}^{\circ}}$ and $\Delta \varphi =0.276{}^{{}^{\circ}}$ for the BESSY II measurements and $\Delta \theta =0.004{}^{{}^{\circ}}$ and $\Delta \varphi =0.360{}^{{}^{\circ}}$ for the laboratory measurements to account for the angular divergence effects in the experimental data. To estimate this broadening for the BESSY II measurements, a Gaussian footprint of 1 mm × 1 mm is assumed. For the laboratory measurements, the assumed footprint size is 0.9 mm × 0.15 mm, with the spot being elongated in the beam direction. From this, 2500 positions are sampled and for each position, emission angles θ and φ are calculated for every center of an area of 8 × 8 pixels on the CMOS detector between $0{}^{{}^{\circ}}<\theta <6{}^{{}^{\circ}}$ and $-3{}^{{}^{\circ}}<\varphi <3{}^{{}^{\circ}}$. The standard deviations of the 2500 angle values for φ and θ in each 8 × 8 pixels area are then averaged to gain a final value for the angular broadening for the pixels in the area. The values for all pixel areas are averaged to produce a mean value. This value is used for the Gaussian broadening. The resulting maps then can be compared to the measurement data. For comparison, an untreated and unstructured ${\mathrm{Si}}_{3}N_{4}$ sample without the grating structure is simulated as well. The fluorescence profile for the unstructured sample is calculated using the approach of Urbach and de Bokx [14]. The sample model consists of a 90 nm ${\mathrm{Si}}_{3}N_{4}$ layer on top of a Si substrate and 3.5 nm ${\mathrm{SiO}}_{2}$ as the top layer.

## 3. Results

The θ-φ-maps for the N Kα and O Kα radiation are shown in Figure 3 for the BESSY II measurement and compared to the simulated data. The results for the LPP measurements are discussed later on. The simulated θ-φ-maps for a sample without the grating structure are shown on top for comparison. As can be seen, they feature no intensity modulation in φ-direction. Below the θ-φ-maps, the θ-profiles at φ = $-0.02{}^{{}^{\circ}}$ are depicted together with the simulated profiles, once with the calculated angular resolution due to the extended footprint (black), once smeared with a Gaussian to approximately match the resolution of the measured data (red, dashed). The experimental profiles for the BESSY II measurements are gained using Specfit as described above, the profiles from the laboratory measurements via ROI evaluation. The ROI profiles feature a width in φ-direction of $0.04{}^{{}^{\circ}}$, while the Specfit profiles have a width of $0.06{}^{{}^{\circ}}$. The simulated intensities are normalized to the measurement data by normalizing the θ-profiles at φ = $-0.02{}^{{}^{\circ}}$ to the maximum value.

In the bottom part of Figure 3, the O Kα interference pattern with several features can be seen. The interference peak at $\theta =1.18{}^{{}^{\circ}}$ and $\varphi =0{}^{{}^{\circ}}$ is visible in the θ-φ-map (red ellipse), as well as the broader peak at about $\theta =3.3{}^{{}^{\circ}}$. The peak at $\theta =1.18{}^{{}^{\circ}}$ is the result of a first waveguide node forming in the XSW field in between the grating lines. The evanescent E-field of this node then reaches far into the ${\mathrm{SiO}}_{2}$ in the upper grating layer. A high intensity of the XSW field corresponds to a high emission probability of a fluorescence photon from that particular place in the sample, leading to the peak in fluorescence intensity. Further fluorescence peaks occur when multiple waveguide nodes form between the grating lines. The broader peak at $\theta =3.3{}^{{}^{\circ}}$ is also visible in the unstructured sample. It originates from interference of the O Kα photons between the interfaces of the ${\mathrm{SiO}}_{2}$ layers, mainly from the top layer. The O Kα signal flattens at about $\theta =5{}^{{}^{\circ}}$, since the path length through the ${\mathrm{SiO}}_{2}$ is then short enough, so that self-absorption of the photons is negligible and all O atoms contribute to the fluorescence emission. The circular shape of the interference features is due to the penetration of the XSW waveguide nodes into the grating lines in dependence on θ and φ. Some radiation damage artifacts of the CMOS chip from previous measurements are visible on the bottom of the map at $\varphi =-1.9{}^{{}^{\circ}}$ (blue ellipse). The N Kα signal is not as clear, since the interference features are damped, but still a faint interference pattern can be seen. The physical processes behind the interference pattern structure of the N Kα signal are the same as for the O Kα signal, but the features occur at different angular positions due to the different wavelength of the fluorescence photons and the different places of origin within the sample. The N Kα signal does not feature a broader peak like the O Kα signal at $\theta =3.3{}^{{}^{\circ}}$, since the distance between the interfaces of the ${\mathrm{Si}}_{3}N_{4}$ structure is too large. Furthermore, since the ${\mathrm{Si}}_{3}N_{4}$ on average is thicker than the ${\mathrm{SiO}}_{2}$ layer, the fluorescence signal does not flatten at a certain value of θ, but instead increases steadily up to the end of the θ-scale.

The interference patterns can be investigated in detail by plotting θ-profiles at φ-values of interest. For this, profiles at $\varphi =-0.02{}^{{}^{\circ}}$ for O Kα and N Kα are shown at the bottom of Figure 3 together with the respective simulations. The black line illustrates the simulation applying the theoretical broadening due to the extended footprint. Since this seems not to describe the measured angular resolution, it is estimated to be $\Delta \theta =\Delta \varphi =0.1{}^{{}^{\circ}}$ for the O Kα line and $\Delta \theta =\Delta \varphi =0.2{}^{{}^{\circ}}$ for N Kα line, assuming an equal broadening in both angular directions. The measured profiles broadened by this estimated angular resolution are depicted by the red dashed profiles. Not only is the feature in the experimental data damped for both fluorescence lines, but the overall intensities in the proximity of the interference features are also reduced. This, as well as part of the increased broadening, is most likely caused by the already observed laterally inhomogeneous carboneous contamination on top of the nanostructure [13]. Additionally, an increase of the carbon signal during the measurements could be observed, indicating insufficient vacuum conditions (∼5 × 10^−5^ mbar). The resulting contamination layer probably not only contains carbon, but other elements as well, mainly oxygen. Therefore, the influence of the contamination on the N Kα and O Kα signal is difficult to quantify. Another influence probably is the large excitation spot of about 1 mm FWHM in the direction perpendicular to the grating lines with a width of also 1 mm. In this way, the unstructured part of the sample is illuminated as well. This leads to a smearing of the interference pattern with the fluorescence signal of the unstructured sample. Note that an ROI evaluation is prone to further angular smearing due to peak overlap; this problem is showcased in Figure 4. However, this effect is negligible here due to the Specfit peak fitting approach.

Nonetheless, the general shapes of the angular profiles are in good agreement with the simulations and thereby prove the general feasibility of the concept. It is demonstrated that the SF-GEXRF technique is applicable in the soft X-ray range and, therefore, for the investigation of ordered nanostructures containing oxides and nitrides. For future measurements, using a focused excitation beam should improve the angular resolution, while the carbon contamination during the measurement can be significantly reduced by improving the vacuum conditions.

In the following, the results of the laboratory measurements at the BLiX will be discussed. Figure 5 shows the experimental θ-φ-maps of O Kα and N Kα in comparison with the respective simulations. The evaluation process is the same as for the BESSY II measurements, except the θ-profiles are directly computed using ROIs instead of spectral fitting, since the low statistics lead to artifacts in the latter approach. A Gaussian excitation footprint of 0.15mm × 0.9mm is assumed, caused by the shallow incidence of the excitation beam on the sample mounted on the 10°-wedge. This leads to an angular broadening of $\Delta \theta =0.002{}^{{}^{\circ}}$ and $\Delta \varphi =0.153{}^{{}^{\circ}}$, with which the simulated data are convolved. Again, this broadening is not sufficient to describe the measured angular resolution and a second empirical broadening is also shown.

The resulting fluorescence patterns of the laboratory measurements are in good agreement with the results from the BESSY II measurements and the simulations. Compared with the BESSY II measurements, the photon maps of the laboratory measurements suffer from lower overall counts due to the lower flux of the LPP source.

The damping of the interference features at $\theta =1.18{}^{{}^{\circ}}$ for O Kα and $\theta =1.58{}^{{}^{\circ}}$ for N Kα is also present here. Damping effects due to peak overlap should be more pronounced in this measurement compared to the synchrotron radiation facility measurements, since an evaluation via Specfit was not feasible. The carbon contamination present should contribute to the damping as well. Additionally, a partial excitation of the unstructured part of the sample is likely. In conclusion, the experimental θ-profiles and θ-φ-maps are in good qualitative agreement with the simulated data. Thus, soft X-ray SF-GEXRF analysis on 2D nanostructures in principle is possible both with synchrotron radiation as well as with a laboratory source.

## 4. Discussion

Scan-free GEXRF measurements of a ${\mathrm{Si}}_{3}N_{4}$ grating in the soft X-ray range at the BESSY II synchrotron radiation facility and at the BLiX laboratory with an LPP are presented. The fluorescence patterns recorded in these proof-of-concept measurements are analyzed via comparison with simulated data gained with finite-element analysis and a Maxwell solver. Utilizing a cost-effective CMOS detector with low readout noise instead of a common CCD enables the discrimination of fluorescence lines below 1 keV. The overall measurement time is also drastically reduced by virtually omitting the readout time, which is especially important when many frames with low exposure time have to be acquired. Thus, by using a CMOS detector instead of a conventional CCD detector, SF-GEXRF investigations in the soft X-ray range are enabled. SF-GEXRF allows scan-free acquisition of the whole range of fluorescence emission angles of interest, while not putting any restrictions on the excitation channel. In this way, neither photon flux, nor excitation spot size has to be compromised.

While SF-GEXRF can easily be implemented at large-scale synchrotron radiation facilities, with improvements to the current setup, even routine measurements in the laboratory may be considered. In the presented measurements, the fluorescence flux of the synchrotron radiation facility setup exceeds the one of the LPP setup by a factor of roughly 100. For the same statistics achieved at the synchrotron radiation facility, the measurement time with the current setup amounts to several days. Future investigations, therefore, could focus on the fitting of sample parameters of interest to measurement results with limited statistics, to gain an estimate of the statistics needed for a reliable characterization of the sample. Nonetheless, improvements to the laboratory setup are feasible. These include the excitation channel, which can be optimized with a larger solid angle of acceptance for the focusing optics or an LPP with an intensity maximum specifically designed for the excitation of the elements of interest. An increase in repetition rate or pulse energy would also be beneficial to achieve an increased photon flux and shorter measurement times. Alternatively, an Al X-ray tube might be utilized. It offers the advantage of a more compact and simplified setup and a comparable resulting fluorescence photon flux as the LPP used in this work [34]. The excitation spot size would roughly stay the same. Since this X-ray tube is commercially available, it is easy to utilize in a purpose-built setup. The currently used Tucsen Dhyana 95 detector also leaves some room for improvement regarding the quantum efficiency below 1 keV [25]. Newer models with improved quantum efficiency could help reduce the measurement time even further [24,35].

In conclusion, scan-free GEXRF offers the ability to analyze 2D nanostructures containing even light elements in the soft X-ray range. With dedicated setups at synchrotron radiation facilities and the expected advances a specialized laboratory setup could offer, routine investigations for quality control or process engineering might become feasible. It might further be utilized for the development of structures such as nanowires [36], metamaterials [37] or light-trapping nanomaterials [38]. Its ability to gain structural ensemble information while being element-sensitive and non-destructive renders SF-GEXRF a useful technique for this purpose, especially when used complementarily with more established approaches.

## Figures and Tables

**Figure 1 nanomaterials-12-03766-f001:**
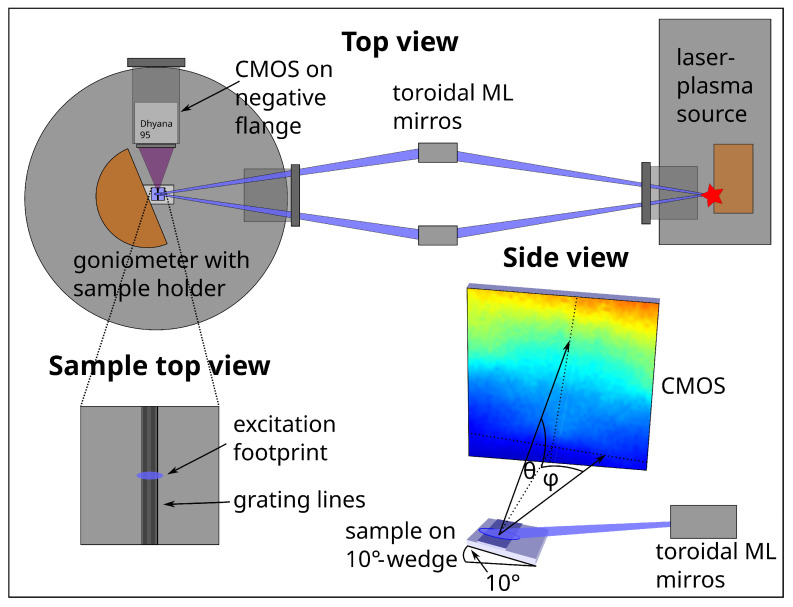
Schematic setup of the laboratory LPP measurement. The LPP vacuum chamber is depicted in the upper right corner of the top view of the setup as the source of soft X-ray radiation, the orange rectangle depicting the Cu target. The pair of toroidal multilayer (ML) optics is located between the LPP and the sample chamber. Note that both chambers and the optics are all kept under vacuum. In the sample chamber on the left, the sample is mounted on a 10${}^{{}^{\circ}}$-wedge on a goniometer (in orange) for alignment and manipulation. The CMOS detector in white is mounted to an inwards-ranging negative flange. In the bottom right, a 3D side view of the sample on the 10${}^{{}^{\circ}}$ wedge with θ and φ and the detector are shown in detail, together with the exciting radiation focused by the ML optics. An exemplary interference pattern is depicted on the CMOS. On the bottom left, a schematic top view of the sample with the grating structure depicted as black lines and the excitation footprint in blue is provided.

**Figure 2 nanomaterials-12-03766-f002:**
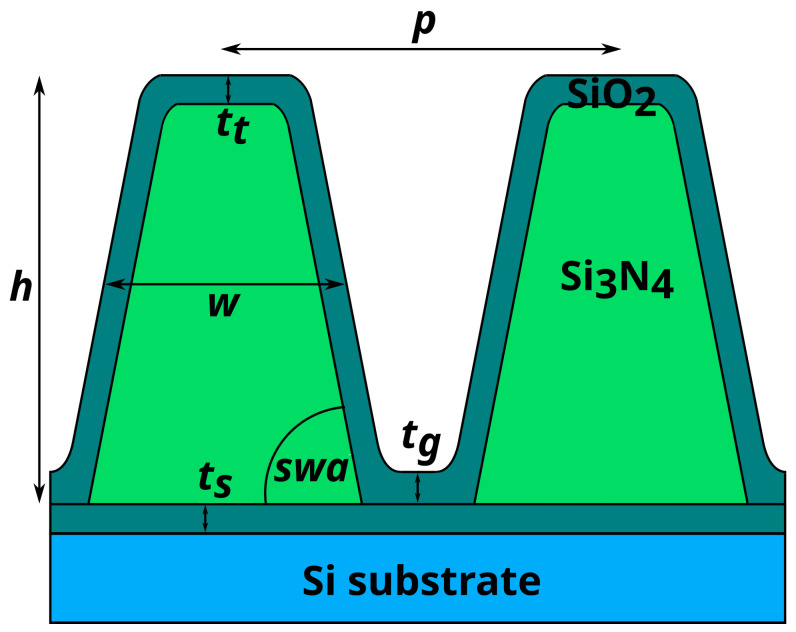
Schematic drawing of the ${\mathrm{Si}}_{3}N_{4}$ sample. The figure depicts the cross section of two grating lines with the sample parameters used for the simulation.

**Figure 3 nanomaterials-12-03766-f003:**
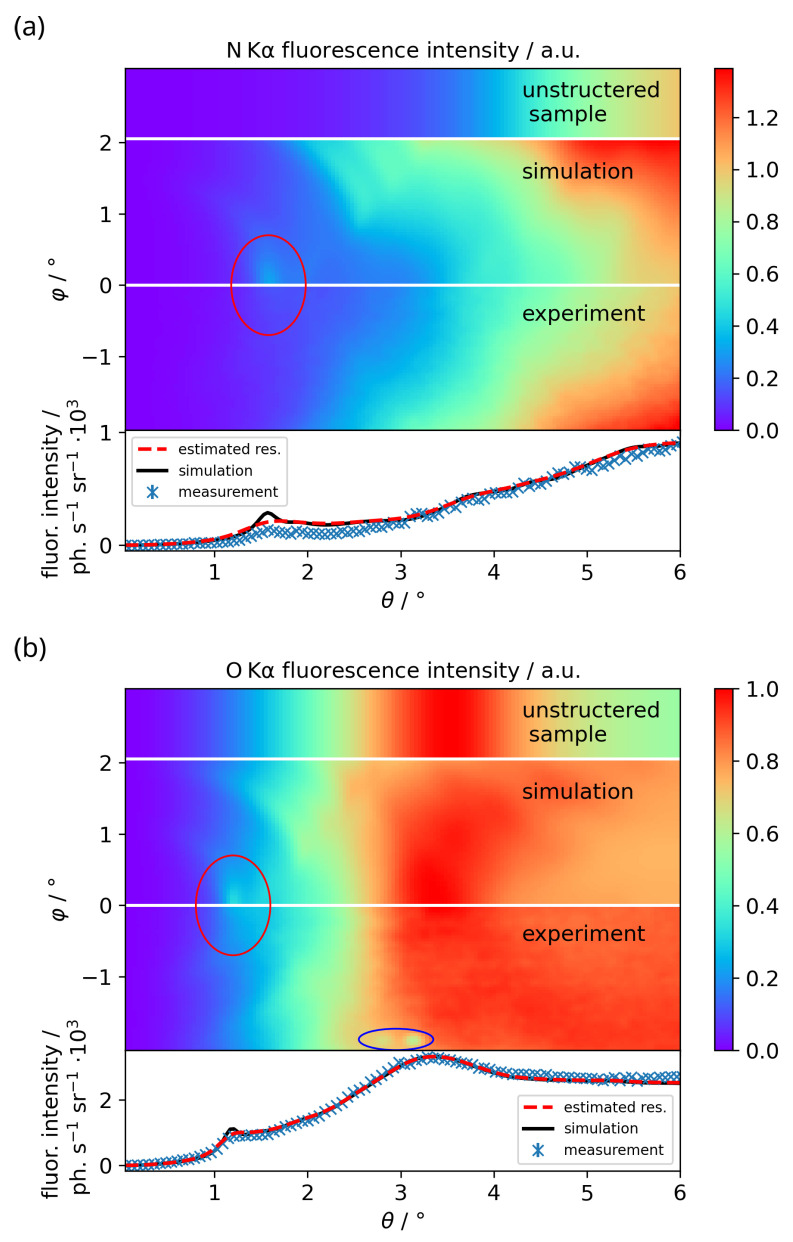
Measured and simulated θ-φ-maps and θ-profiles at φ = $-0.02{}^{{}^{\circ}}$ for (**a**) N Kα on top and (**b**) O Kα below from the ${\mathrm{Si}}_{3}N_{4}$ lamellar grating sample from the SF-GEXRF BESSY II measurement. The top parts of the maps show the θ-φ-maps for an unstructured sample with 90 nm ${\mathrm{Si}}_{3}N_{4}$ on top of a silicon substrate and 3.5 nm of ${\mathrm{SiO}}_{2}$ as the top layer. Below are shown the simulated θ-φ-maps and the bottom θ-φ-maps depict the measured data. The experimental θ-φ-maps are smoothed with a Gaussian with an FWHM of $0.04{}^{{}^{\circ}}$ for presentation. The maps are normalized with the respective θ-profiles at φ = $-0.02{}^{{}^{\circ}}$. The red ellipse highlights the interference feature between θ = 1° and θ = 2°. Some radiation damage artifacts of the CMOS chip from previous measurements are visible at the bottom of the O Kα map, marked by a blue ellipse. The blue θ-profiles at the bottom depict the measurement data obtained from the Specfit evaluation. The error bars denote the statistical uncertainties. The black solid line denotes the simulation broadened by the calculated angular resolution due to the extended excitation footprint. The dashed red profile denotes the simulated profile convolved with a Gaussian with an FWHM of $0.1{}^{{}^{\circ}}$ in θ and φ direction for O Kα and $0.2{}^{{}^{\circ}}$ for N Kα to provide an estimation for the angular resolution achieved in the measurement.

**Figure 4 nanomaterials-12-03766-f004:**
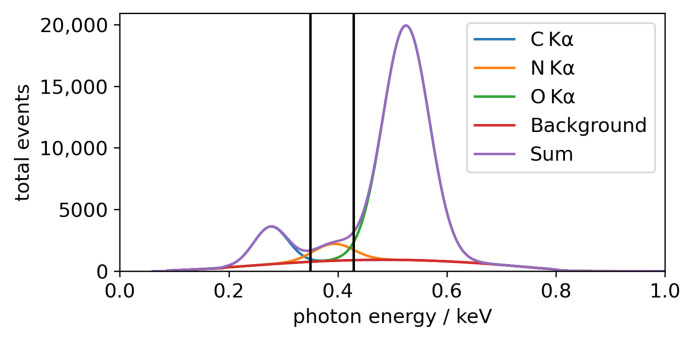
Fit result of the energy-dispersive spectrum of the BESSY II measurement at the area of the N Kα interference pattern at $\theta =1.58{}^{{}^{\circ}}$ and $\varphi =0{}^{{}^{\circ}}$. The C Kα, N Kα and O Kα fluorescence line profiles fitted by Specfit [31] are shown, together with the subtracted background. The sum spectrum is also shown. The black vertical lines denote an exemplary ROI from 350ev to 430ev for N Kα.

**Figure 5 nanomaterials-12-03766-f005:**
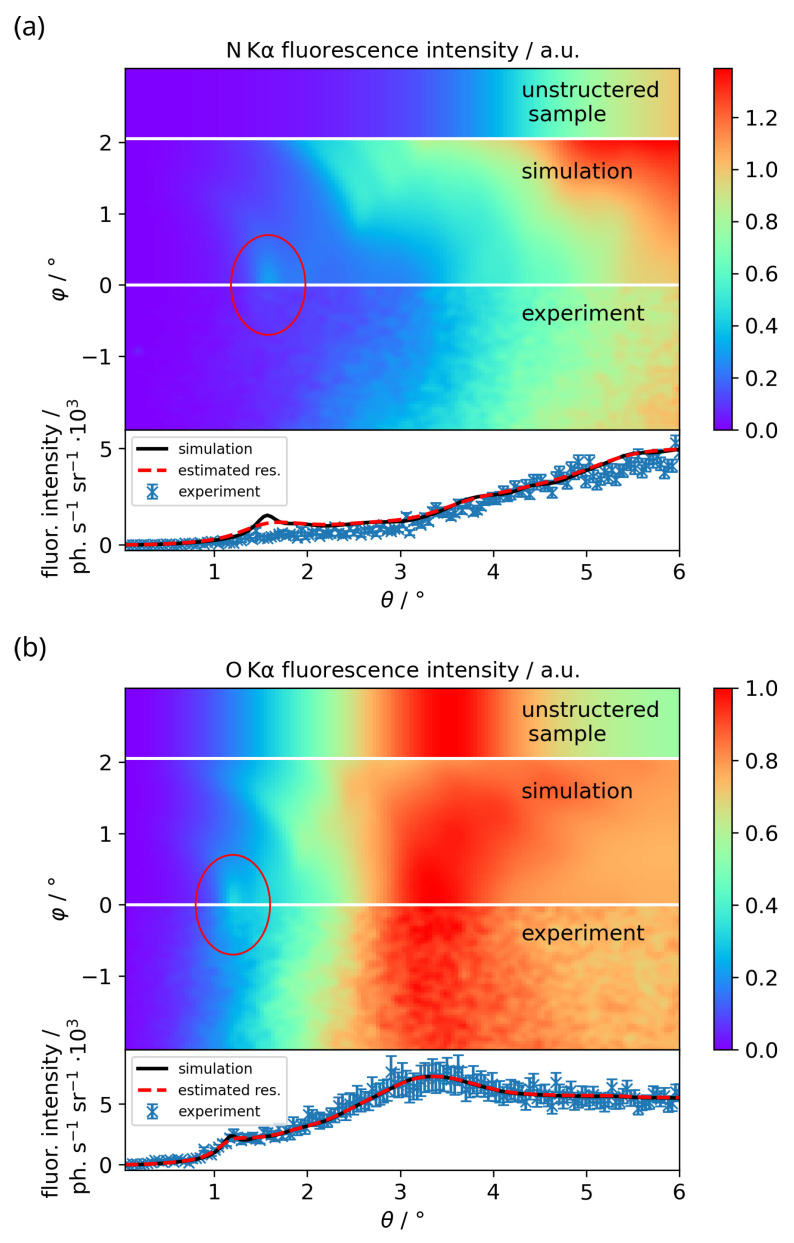
Measured and simulated θ-φ-maps and θ-profiles from the ${\mathrm{Si}}_{3}N_{4}$ lamellar grating recorded with the laboratory SF-GEXRF spectrometer utilizing the LPP source. Notation and figure structure are as in Figure 3. For computing the SF-GEXRF profile at φ = $-0.02{}^{{}^{\circ}}$, the ROI method is used due to low counting statistics.

## Data Availability

The data presented in this study are available on request from the corresponding author.

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
