# Peer review of "Scan-Free GEXRF in the Soft X-ray Range for the Investigation of Structured Nanosamples"

_nanomaterials, 2022, doi:10.3390/nano12213766_

Round 1

Author Response

We like to thank the reviewer for the detailed report.

Reviewer 2 Report

The authors present their work conducted for the measurement of scan free GEXRF in the soft X-ray range at BESSY II synchrotron radiation facility. They frame in detail the radiation line, the laboratory and also the proof-of-concept fluorescence models used for analysis by comparison with simulated data obtained with finite element analysis and a Maxwell solver.

The overall measurement method starting from the acquisition procedures and the criticalities on the dimensions of the excitation spots and the angles of interest of fluorescence emission are well described together with the configuration of the measurement setup.

The work carried out is certainly well planned and executed and offers a method and a specialized laboratory configuration for quality control investigations.

One aspect that needs improvement in the manuscript is the description of the applications or science of nanomaterials to which this study can make a significant scientific new contribution beyond service to process engineering.

Author Response

The following remarks have been addressed:

"One aspect that needs improvement in the manuscript is the description of the applications or science of nanomaterials to which this study can make a significant scientific new contribution beyond service to process engineering."

Further application examples of SF-GEXRF are mentioned now in the introduction (lines 66-70). More possibilites of the application of SF-GEXRF for the characterization and development of nanomaterials are mentioned at the end of the conclusion.

We like to thank the reviewer for the detailed report.

Reviewer 3 Report

The paper devoted to development of the new progressive method for studying structured nanosamples, namely the Scan-free grazing-emission X-ray fluorescence spectroscopy (SF-GEXRF). The method itself and the original results presented in the paper are very interesting and have a significant impact on subsequent applications.

However, the presentation of the results is not as impressive as it should be. The main line of presentation consists in numerous technical details of the experiment carried out at the BESSY synchrotron and in the laboratory with CMOS detector and a laser-produced plasma source (LPP). It is clear that the authors had to overcome a huge number of technical problems for obtaining the real results, and the right choice of the using appliances were crucial. At the same time for the readers of Nanomaterials the physical results concerning the nanostructure which have been obtained or can be obtained in the future are of particular interest. About that point I have found only two sentences in the whole article:

The Si3N4 lamellar grating is well-known from previous studies and the parameters of the grating are published in [13]. It has been characterized and a validated model exists.

Oxygen was found on top of the grating structure, probably contained in a 3.5nm thick SiO2 layer [10].

It is written that

The results are compared to calculations of the sample model performed by a Maxwell solver based on the finite-elements method.  But what was the model? The readers want picture and explanation how lamellar grating enter to the calculations. To what feature of the sample does the interference spot refer? And what is the physical meaning of the difference in the pictures for O and N.

Another point is that the illustration in Fig.1 is too complicated for understanding. To be honest, I had to go back to previous articles to understand the meaning of angles. CMOS is orange or grey? The beam footprint is gray or small blue square? In the bottom half, a side view of the sample and beam is in the opposite direction than in the upper part.

In total, it is wonderful that the authors confirm the possibility to use the method in the Lab (I would not say that the method is well established) and the article has great interest for reader, but the style of presentation should contain more physics of interaction instead of the technical details.

Author Response

The following remarks have been addressed:

"At the same time for the readers of Nanomaterials the physical results concerning the nanostructure which have been obtained or can be obtained in the future are of particular interest."

" But what was the model? The readers want picture and explanation how lamellar grating enter to the calculations. To what feature of the sample does the interference spot refer? And what is the physical meaning of the difference in the pictures for O and N."

A schematic cross-section drawing of the sample was added including the parameters used in the simulation (Figure 2). The parameters used in the model for the simulation and more possible parameters that could be gained from fitting the experimental data are discussed briefly (lines 183-191). The physical origins of the interference features in the O and N patterns and their differences are explained now (lines 228-240).

"Another point is that the illustration in Fig.1 is too complicated for understanding. To be honest, I had to go back to previous articles to understand the meaning of angles. CMOS is orange or grey? The beam footprint is gray or small blue square? In the bottom half, a side view of the sample and beam is in the opposite direction than in the upper part."

Sorry for the confusion, Figure 1 now was modified for more clarity. A more elaborate explanation of the theta and phi angles was added (lines 149-153).

We like to thank the reviewer for the detailed report.

Round 2

Reviewer 3 Report

The authors essentially improved the text including the detailed explanation of the measured fluorescence pattern and essentially improved the picture of the experimental scheme.

However, as it seems to me, on the side view the vector phi should be out of the plane of picture (or plane of incidence), now it is only for phi=90o (psi = 90° is the direction perpendicular to the grating lines). In general, the designation of the angles by the vectors but not by the semicircles between two vectors is rather unclear. In the explanation as well the terms “the meridional angle and the sagittal angle” are rather complicated, probably it is simply “the angle with the sample surface” and “the azimuth angle” meaning the deviation from the direction of the grating lines?

Author Response

The following remarks have been adressed:

"However, as it seems to me, on the side view the vector phi should be out of the plane of picture (or plane of incidence), now it is only for phi=90o (psi = 90° is the direction perpendicular to the grating lines). In general, the designation of the angles by the vectors but not by the semicircles between two vectors is rather unclear."

Sorry again for the confusion, to make the definition of the angles more clear, a 3D view of the setup geometry has been included in figure 1.

"In the explanation as well the terms “the meridional angle and the sagittal angle” are rather complicated, probably it is simply “the angle with the sample surface” and “the azimuth angle” meaning the deviation from the direction of the grating lines?"

The suggested description of the angles has been adopted (line 150-151). The same description was subsequently also applied to the explanation of GIXRF in the introduction (line 40-44).

We like to thank the reviewer again for the remarks.